# Discontinuation of alpha-blocker therapy in men with lower urinary tract symptoms: a systematic review and meta-analysis

Henk van der Worp [1], Petra Jellema,[1] Ilse Hordijk,[2] Yvonne Lisman-van Leeuwen,[1] Lisa Korteschiel,[2] Martijn G Steffens,[2] Marco H Blanker [1]

**To cite:** van der Worp H, Jellema P, Hordijk I, *et al.* Discontinuation of alpha-blocker therapy in men with lower urinary tract symptoms: a systematic review and meta-analysis. *BMJ Open* 2019;9:e030405. doi:10.1136/bmjopen-2019-030405

¹Department of General Practice and Elderly Care Medicine, University of Groningen, University Medical Center Groningen, Groningen, The Netherlands
²Urology, Isala Hospitals, Zwolle, The Netherlands

**Correspondence to**
Dr Henk van der Worp;
h.van.der.worp@umcg.nl

## ABSTRACT

**Objectives** We aimed to synthesise the available data for the effect of stopping alpha-blocker therapy among men with lower urinary tract symptoms. The focus was on symptom, uroflowmetry and quality of life outcomes, but we also reviewed the adverse events (AEs) and the number of patients who restarted therapy.

**Data sources** We searched MEDLINE/PubMed, EMBASE/Ovid and The Cochrane Central Register of Controlled Trials from inception to May 2018.

**Eligibility criteria** We selected studies regardless of study design in which men were treated with an alpha-blocker for at least 3 months and in which the effects of alpha-blocker discontinuation were subsequently studied. Only controlled trials were used for the primary objective.

**Data extraction and synthesis** Two reviewers independently extracted data and assessed the risk of bias for the controlled studies only using the Cochrane Collaboration's tool for assessing risk of bias. Data were pooled using random-effects meta-analyses.

**Results** We identified 10 studies (1081 participants) assessing the primary objective. Six studies (733 participants) assessed differences in AEs between continuation and discontinuation, and six studies (501 participants) reported the numbers of subjects that restarted treatment after discontinuation. No studies in primary care were identified. After discontinuing monotherapy, symptom scores increased and peak flow rates decreased at 3 and 6 months, but not at 12 months; however, neither parameter changed when alpha-blockers were stopped during combination therapy. Small differences in post-void residual volumes and quality of life scores were considered clinically irrelevant. We also found that 0%–49% of patients restarted after stopping alpha-blocker therapy and that AEs did not increase with discontinuation.

**Conclusions** Discontinuing alpha-blocker monotherapy leads to a worsening compared with continuing therapy. Discontinuing the alpha-blocker after combination therapy had no significant effects on outcomes in either the short or long term. Discontinuation may be appropriate for the frail, elderly or those with concomitant illness or polypharmacy. However, studies in primary care are lacking.

**PROSPERO registration number** CRD42016032648.

## Strengths and limitations of this study

► This is the first systematic review that synthesises the literature concerning alpha-blocker discontinuation.
► The review was conducted in accordance with the Preferred Reporting Items for Systematic Reviews and Meta-Analyses guidelines.
► The number of studies that could be included was limited, and the risk of bias was high for most outcomes preventing the drawing of firm conclusions.

## INTRODUCTION

Alpha-blockers are the first-choice treatment for men with moderate-to-severe lower urinary tract symptoms (LUTS) because of their proven, but small, superiority over placebo,[1–3] but their use can be associated with dizziness, orthostatic hypotension and increased fall risk.[3 4] This may be especially problematic in the elderly, who often have polypharmacy and multimorbidity. Given the natural course of LUTS, with 30% of patients showing improvement over time,[5] it may be appropriate to consider discontinuation of alpha-blocker therapy, especially in the elderly. There are no clear data on the effect of this approach but the guideline on male LUTS for Dutch general practitioners (GPs) advises that alpha-blocker therapy be discontinued after 3–6 months, followed by symptom review.[2] By contrast, guidelines followed by urologists do not advocate routine discontinuation,[1 6] although the European Association of Urology (EAU) do mention that alpha-blocker discontinuation may be considered after 6 months in the context of combination therapy.[1]

A number of researchers have studied the effects of discontinuing alpha-blockers, but to date, there has been no synthesis of this literature.

We performed a systematic review and meta-analysis to obtain data about the effect of discontinuing alpha-blockers on male LUTS. Our primary objective was to compare the effects of discontinuing therapy with those of continuing therapy. Secondary objectives were (1) to determine the proportion of men who restarted alpha-blocker therapy and (2) to determine the possible adverse effects (AEs) of both discontinuation and continuation.

## METHODS

We completed this review according to the Preferred Reporting Items for Systematic Reviews and Meta-Analyses guidelines and registered the protocol in the PROSPERO database (CRD42016032648) (http://www.crd.york.ac.uk/PROSPERO/display_record.php?ID=CRD42016032648).

### Selection criteria

We selected studies in which men were treated with an alpha-blocker for at least 3 months and in which the effects of alpha-blocker discontinuation were subsequently studied. For the primary objective, only randomised controlled trials (RCTs, including quasi-randomised trials) and non-randomised trials (NRTs) that compared alpha-blocker discontinuation to continuation were selected. For the secondary objectives, we also included uncontrolled studies. At all stages, we excluded studies written in languages other than Dutch, English, French or German.

### Outcome measures

The following outcomes were used for the primary objective: symptom scores, such as the International Prostate Symptom Score (IPSS); urinary flow rates; post-void residual urine volume (PVR); and quality of life (QoL). For the secondary objectives, we calculated the percentage of patients who restarted alpha-blocker therapy and the numbers of AEs in the continuation and discontinuation groups.

### Search methods for identification of studies

We searched MEDLINE/PubMed, EMBASE/Ovid and The Cochrane Central Register of Controlled Trials using search terms covering LUTS, alpha-blockers and discontinuation (see online supplementary file 1 for detailed information). We ran the searches in January 2016 and updated them in July 2017 and May 2018. The reference lists of relevant articles were also screened to identify additional eligible studies. All duplicate files were removed before the titles and abstracts of the remaining records were independently screened by three reviewers (IH, LK, MHB) and classified as 'inclusion,' 'exclusion,' or 'uncertain.' Next, the same reviewers independently applied the selection criteria to the full-text papers of all records classified into the inclusion or uncertain groups, and decided whether to include or exclude the research.

Discrepancies in the selection procedure were resolved by consensus.

### Data extraction process

Two authors (HvdW, IH) independently performed data extraction using standardised forms. We extracted the following data: (1) the participant characteristics; (2) the interventions used; (3) the primary and secondary outcomes, as well as the timing of the outcome assessment; and (4) the study design. If possible, we extracted data by allocated intervention to allow an intention-to-treat analysis. Discrepancies were resolved by re-examination and discussion of the full-text papers or by consultation with a third author (MHB).

### Risk of bias

Two reviewers (HvdW, YL-vL) independently assessed the risk of bias using the Cochrane Collaboration's tool for assessing risk of bias.[7] This tool includes six domains, as follows: selection bias, performance bias, detection bias, attrition bias, reporting bias and other bias. Only RCTs and NRTs were assessed because these were used for the primary objective. Discrepancies in the risk of bias assessment were resolved by consensus or arbitration with a third party (MHB). Risk of bias was described for the five domains (selection, performance, attrition, reporting and other) and summarised across studies and outcomes.[7] To ascertain graphically the existence of publication bias, the construction of funnel plots was planned in case at least 10 studies were included.

### Data analysis

Data were analysed using Review Manager, V.5.3.[8] We calculated the risk difference and 95% CIs for dichotomous variables (inverse-variance method) and the mean differences (MD) with 95% CIs for continuous variables (Mantel-Haenszel method). For AEs, we also calculated the rate of AEs per 1000 patient-days based on the sample sizes and follow-up times.

Clinical heterogeneity was assessed by checking the characteristics of participants and interventions. Statistical heterogeneity was assessed by visual inspection of the forest plots and of the results of statistical testing for heterogeneity ($I^2$ statistic). We pooled data if we identified two or more studies with an $I^2$ of <40%,[9] using a random effects model. Data from both RCTs and NRTs were pooled. Synthesising and pooling were done separately for monotherapy and combination therapy. If the data for pooling were only presented in figures (eg, SD), it was extracted from those figures. If data were not present at all in the article, we contacted the authors if the article had been published in the past 10 years. If data could not be obtained in this way, we imputed data from a previous meta-analysis on the efficacy of alpha-blockers,[3] as described in the Cochrane Handbook.[9 10]

The following cut-off values were used to define the minimal clinical important difference (MCID): 2.7 points for IPSS,[11] 2 mL/s for Q-max[3] and 0.5 point for

the IPSS-QoL scores.[3] The MCID is the smallest change in a treatment outcome that an individual patient would identify as important and which would indicate a change in the patient's management.

## Patient and public involvement

This study was performed without patient involvement. The patients were not invited to comment on the study design and were not consulted to interpret the results. The patients were not invited to contribute to the writing or editing of this document for readability or accuracy.

## RESULTS

The searches yielded 1039 publications (online supplementary file 2), of which 16 with a total of 1823 participants were included (table 1). All included studies were performed in secondary or tertiary care. Nine studies (772 participants) reported discontinuing alpha-blocker monotherapy: two double-blind RCTs,[12 13] two open-label RCTs,[14 15] one NRT[16] and four uncontrolled studies.[17–20] Six studies (980 participants) reported discontinuing alpha-blockers used in combination therapy: one double-blind RCT,[21] four open-label RCTs[22–25] and one uncontrolled study.[26] Finally, one uncontrolled study (n=71) reported discontinuing both alpha-blocker monotherapy and combination therapy.[27]

Two of the included studies randomised patients into three groups: a discontinuation group, a continuation group and a third group that continued with alternate-day use of alpha-blockers.[14 15] We only used the data from the discontinuation and continuation groups.

In another three studies, the data required for pooling were missing.[12 13 21] Because these studies had been published over 15 years previously, no efforts were made to contact the authors. For one of the studies, means and SD could be obtained from the figures.[13] For the other two studies,[12 21] the SD were missing and were imputed from the results of a previous meta-analysis (see online supplementary file 3).[3] No studies were excluded form pooling based on statistical heterogeneity.

## Risk of bias

Most of the included studies had risks of bias (online supplementary file 4). The most common were lack of blinding and randomisation, with only 3 out of 10 studies having a 'low risk' for these items. There was no evidence of reporting bias in any of the included studies. The summary of bias by study indicated that only one study had low risk of bias,[12] while two studies had unclear risks of bias,[13 21] and the remaining studies had high risks of bias.[14–16 22–25] As a result, risk of bias was high for all but one outcome.

We did not construct funnel plots to ascertain the existence of publication bias graphically, as the number of included studies in each meta-analysis was less than 10.

## Effects of alpha-blocker discontinuation

Five studies of monotherapy (n=341)[12 13 15 16] and five studies of combination therapy (n=740)[21–25] were used for the primary research objective. Only three provided data on the number of patients who did not comply with the intervention and who restarted alpha-blocker use after discontinuation. Two of these provided a per protocol analysis excluding those patients[24 25] and the third provided an intention-to-treat analysis for categorised variables, with a per protocol analysis for the raw outcomes,[23] effectively precluding an intention-to-treat analysis. Because all included studies reported outcomes at 3, 6 or 12 months, we compared outcomes at these time points.

### Symptom scores

All but one study[13] assessed symptoms with the IPSS questionnaire, or its predecessor the American Urological Association (AUA) symptom score (n=1054).

By 3 months after discontinuing alpha-blocker monotherapy, symptoms increased in the discontinuation group compared with the continuation group (MD=4.17; 95% CI 2.91 to 5.43),[14 15] whereas there was no difference between the continuation and discontinuation groups in the studies of combination therapy (MD=0.97; 95% CI −0.32 to 2.27, figure 1A).[21 25]

After 6 months, two RCTs and one NRT on monotherapy found a significant worsening of symptoms (differences varying from 2.0 to 5.8 points) in subjects that discontinued alpha-blockers.[12 14 15] No difference was found for studies on combination therapy after 6 months (MD=0.56; 95% CI −1.57;2.69, figure 1B).[24 25]

After 12 months, the one study that looked at discontinuing monotherapy found a non-significant difference of 1.2 points between groups.[16] No differences were found in two open-label RCTs that looked at alpha-blocker discontinuation after combination therapy.[22 23] Another NRT presented data for both groups, but did not make a direct statistical comparison between groups.[25] Data could not be pooled ($I^2$=61%).

### Peak urine flow rate

Nine studies (804 patients) assessed peak urine flow rate (Q-max): five studies for monotherapy,[12–16] and four studies for combination therapy.[22–25]

After 3 months, Q-max was reduced by 2.59 mL/s (95% CI 1.40 to 3.77, figure 2A) in those who discontinued alpha-blocker monotherapy compared with those who continued therapy.[13–15] A single study on discontinuing combination therapy found that there was an increase of 1.4 mL/s in those who discontinued compared with those who continued therapy, but the researchers did not perform a statistical comparison.[25]

After 6 months, a reduction was again found in the Q-max after discontinuing monotherapy (MD=1.79; 95% CI 0.73 to 2.86),[12 14 15] but no difference was found after

**Table 1** Characteristics of the controlled and uncontrolled studies of monotherapy and combination therapy

| Authors | Design | Type of AB | Daily dose | AB use (Before stopping) | No. of patients (Stop phase) | Age (Mean) | IPSS at baseline (Mean) | Measured outcomes Primary | Secondary | Follow-up (months) |
|---|---|---|---|---|---|---|---|---|---|---|
| **Controlled trials** | | | | | | | | | | |
| *Monotherapy* | | | | | | | | | | |
| Fabricius et al[13] | Double-blind RCT | TER | 10 mg | 24 weeks | 27 | 68 | – | Q-max, Q-avg, PVR, | AEs | 3 |
| Debruyne et al[12] | Double-blind RCT | TER | 5 mg/10 mg | 26 weeks | 167 | 63.6 | 19.1 | IPSS, Q-max, QoL | AEs | 6 |
| Gerber et al[16] | NRT | DOX | 4 mg | 3 months | 37 | 65 | C=20.9; DC=21.5 | IPSS | Restart | 12 |
| Kaplan et al[15] | Open-label RCT (quasi-randomised) | ALF | 7.5 mg | 3 months | 53 | 60.5 | 15.6 | IPSS, Q-max | AEs | 3/6 |
| Yanardag et al[14] | Open-label RCT | TAM | 0.4 mg | 3 months | 57 | 61.3 | 12.3 | IPSS, Q-max, Q-avg, PVR | AEs | 3/6 |
| **Combination therapy** | | | | | | | | | | |
| Barkin et al[21] | Double-blind RCT | TAM | 0.4 mg | 24 weeks | 277 | C=67.6; DC=66.9 | C=16.4; DC=16.5 | IPSS, QoL | AEs | 3 |
| Liaw and Kuo[25] | Open-label RCT (quasi-randomised) | TAM | 0.2–0.4 mg | 1 year | 47 | C=70.7; DC=72.1 | 15.6 | IPSS, Q-max, QoL | Restart | 3/6/12 |
| Lee et al[24] | Open-label RCT | TAM | 0.2 mg | 48 weeks | 69 | 68 | 15.3 | IPSS, Q-max, QoL | Restart, AEs | 6 |
| Lin et al[23] | Open-label RCT | DOX | 4 mg | 2 years | 230 | 75 | C=13.1; DC=15.6 | IPSS, Q-max, PVR | Restart | 12 |
| Matsukawa et al[22] | Open-label RCT | SIL | 8 mg | 12 months | 117 | C=70.1; DC=69.1 | C=17.4; DC=17.2 | IPSS, Q-max, PVR, QoL | AEs | 12 |
| **Uncontrolled studies** | | | | | | | | | | |
| *Monotherapy* | | | | | | | | | | |
| Kobayashi et al[18] | CS | TAM | 0.2 mg | 28.5±26.8 months | 33 | 70.4 | 16.3 | | Restart | 6 |
| Yokoyama et al[20] | CS | NAF/ TAM / URA | 25–50 mg/0.2 mg/30 mg | 2–200 months | 60 | 70 (median) | 15.9 | | Restart | 12 |

Continued

**Table 1** Continued

| Authors | Design | Type of AB | Daily dose | AB use (Before stopping) | No. of patients (Stop phase) | Age (Mean) | IPSS at baseline (Mean) | Measured outcomes Primary | Measured outcomes Secondary | Follow-up (months) |
|---|---|---|---|---|---|---|---|---|---|---|
| Nickel et al[19] | CS | ALF/DOX/TAM/TER | No data | 9 months | 220 | 66.1 (total sample) | 19.9 | | Restart | 9 |
| Chung et al[17] | CS | ALF | 10 mg | 12 weeks | 58 | 68.6 (total sample) | 16.7 (total sample) | | Restart | 6 |
| Combination therapy | | | | | | | | | | |
| Baldwin et al[26] | CS | DOX | 2–8 mg | 3–12 months | 240 | 66 (total sample) | Range: 20–33 (total study sample) | | Restart | 1 |
| Both* | | | | | | | | | | |
| Kuo[27] | CS | DIB | 20 mg | 6 months | ABM=71; ABC=65 | ABM=66.3; ABC=66.8 | ABM=21.2; ABC=22.5 | | Restart | 1 |

*Both monotherapy and combination therapy discontinuation.
AB, alpha-blocker; ABC, alpha-blocker combination treatment; ABM, alpha-blocker monotherapy; AEs, adverse events;ALF, alfuzosin; C, continuation group; CS, cohort study; DC, discontinuation group;DIB, dibenyline; DOX, doxazosin; NAF, naftopidil; NRT, non-randomised controlled trial: ns, not stated; PVR, post-void residual urine volume; Q-avg, average urine flow rate; Q-max, peak urine flow rate; SIL, silodosin; TAM, tamsulosin; TER, terazosin; URA, urapidil.

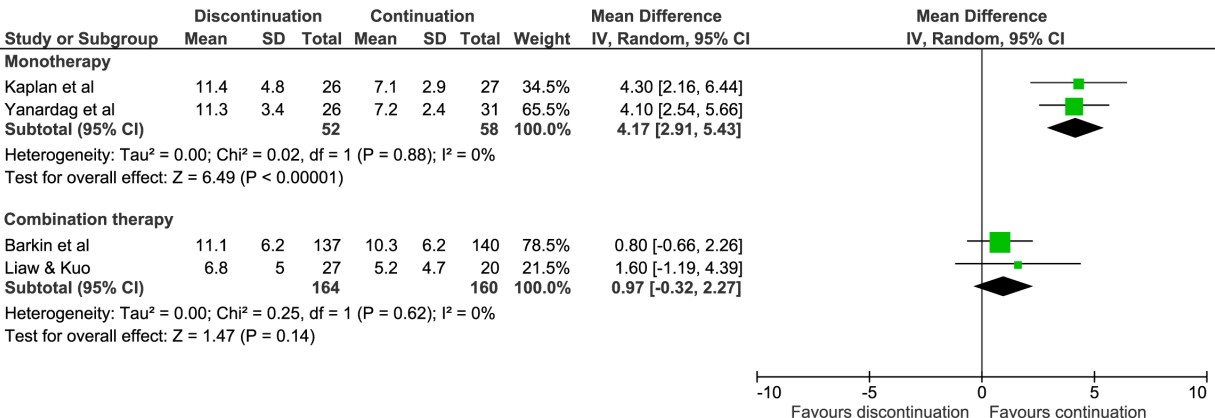

**Figure 1** Forest plots of the IPSS when discontinuing or continuing alpha-blockers. (A) Forest plot of the IPSS after 3 months for alpha-blocker discontinuation or continuation. (B) Forest plot of the IPSS after 6 months for alpha-blocker discontinuation or continuation. IPSS, International Prostate Symptom Score.

discontinuing in the context of combination therapy (MD −0.23; 95% CI −1.51 to 1.05, figure 2B).[24 25]

After 12 months, no differences were found in an NRT reporting on the effects of discontinuing monotherapy.[16] Three studies assessed Q-max 12 months after combination therapy. Among these, two open-label RCTs found no difference between groups[22 23]: one study found a difference of 0.1 mL/s in favour of the continuation group and the other found that 7% fewer patients in the discontinuation group had a reduction in Q-max of >2 mL/s compared with the continuation group. Again, the NRT on combination therapy showed an increase of 2.5 mL/s after alpha-blocker discontinuation, which was not seen in the group that continued alpha-blockers, but differences were not tested.[25] Data could not be pooled ($I^2$=68%).

### Average urine flow rate

Data from two RCTs on monotherapy (84 patients) could not be pooled ($I^2$=70%).[13 14]

After 3 months, one RCT reported a reduction of 2.2 mL/s in subjects who discontinued therapy compared with those who continued therapy,[13] whereas no statistical testing of the difference of 0.6 mL/s between groups was performed in the other RCT.[14]

After 6 months, this second study found a difference of 0.9 mL/s between groups in favour of continuing monotherapy.[14]

### Post-void residual volume

PVR was measured in five studies (n=468), with three measuring it after discontinuing monotherapy[13 14 16] and two measuring it after discontinuing the alpha-blocker in combination therapy.[22 23]

After 3 months, discontinuing monotherapy resulted in a PVR increase of 9.98 mL (95% CI 0.84 to 19.12, figure 3).[13 14]

After 6 months, an open-label trial on discontinuing monotherapy (57 participants) also found a statistically significant difference (14.1 mL) in favour of continuing therapy.[14]

At 12 months after discontinuing monotherapy, an NRT did not find a significant difference between groups.[16] Two open-label RCTs on discontinuing combination therapy did report non-significant differences: one showed a 2 mL difference between groups,[22] and the other showed that 8% more patients in the discontinuation group reported a PVR increase of >50%.[23]

### Quality of life

All five studies (one of monotherapy and four of combination therapy; 677 patients) that assessed QoL used the IPSS QoL subscore.

After 3 months, one study of combination therapy found no difference between groups (0 point).[21] In another NRT of combination therapy, a difference of 0.4 point was reported in favour of the group that discontinued therapy, but this was not statistically tested.[25]

### 2A. Q-max - 3 months

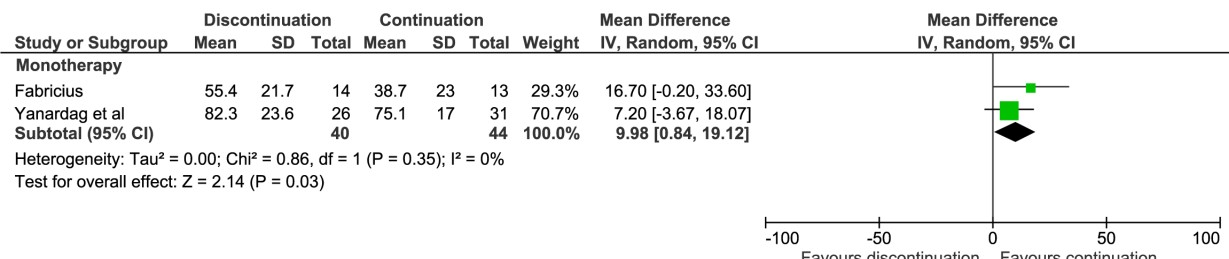

### 2B. Q-max - 6 months

**Figure 2** Forest plots of the Q-max when discontinuing or continuing alpha-blockers. (A) Forest plot of the Q-max 3 months after alpha-blocker discontinuation or continuation. (B) Forest plot of the Q-max 6 months after alpha-blocker discontinuation or continuation. Q-max, peak urine flow rate.

After 6 months, one study on monotherapy found a statistically significant difference of 0.2 point in favour of those who continued alpha-blocker therapy.[12] This was the only outcome with a low risk of bias. A difference was found for the pooled studies of combination therapy (MD=0.42; 95% CI 0.11 to 0.73, figure 4).[24 25]

After 12 months, no differences in QoL scores (only 0.1 point in favour of discontinuation) were found in an RCT of 117 participants receiving combination therapy.[22] Another NRT of patients receiving combination therapy found a difference of 0.4 in favour of continuation, but did not compare groups statistically.[25] Data could not be pooled (I[2]=60%).

### Restart of prior treatment and AEs
#### Patients restarting treatment

Six studies (501 patients) reported data on restarting alpha-blockers after discontinuation.[16 17 20 23–25] In three of these (187 patients), 7%–49% of subjects restarted alpha-blockers after 6 months.[17 20 24] One of these,[20] together with another three studies (374 patients),[16 23 25] described that 0%–33% of subjects had restarted alpha-blockers at 12 months. The highest (49%) and lowest percentages (0%) restarting therapy were found in studies of monotherapy.[16 17] However, four of the included studies explicitly advised subjects to restart alpha-blocker use if their PVR was >100 mL,[17 20] or if

### PVR - 3 months

**Figure 3** Forest plot of the PVR 3 months after discontinuing or continuing alpha-blockers. PVR, post-void residual urine volume.

## QoL - 6 months

| Study or Subgroup | Discontinuation | | | Continuation | | | Weight | Mean Difference IV, Random, 95% CI |
|---|---|---|---|---|---|---|---|---|
| | Mean | SD | Total | Mean | SD | Total | | |
| **Combination therapy** | | | | | | | | |
| Lee et al | 3.1 | 0.1 | 33 | 2.6 | 1.2 | 36 | 61.3% | 0.50 [0.11, 0.89] |
| Liaw & Kuo | 1.85 | 1.04 | 27 | 1.55 | 0.69 | 20 | 38.7% | 0.30 [−0.20, 0.80] |
| **Subtotal (95% CI)** | | | 60 | | | 56 | 100.0% | 0.42 [0.11, 0.73] |

Heterogeneity: Tau² = 0.00; Chi² = 0.38, df = 1 (P = 0.54); I² = 0%
Test for overall effect: Z = 2.69 (P = 0.007)

**Figure 4** Forest plot of the QoL score 6 months after discontinuing or continuing alpha-blockers. QoL, quality of life.

symptoms worsened.[23 25] These studies reported the highest restart rates.

Three other studies provided indirect information about restarting alpha-blocker use.[18 26 27] Two of these reported on successful discontinuation, defined as no increase in symptoms and no request for continuation of treatment. After 6 months, one indicated success among 69% of those receiving monotherapy.[18] Another study reported success rates of 13%–87% 1 month after discontinuing combination therapy, with percentages increasing as the duration of alpha-blocker use increased (ranging from 3 to 12 months).[26] Discontinuation was successful in 13%–20% of subjects who used alpha-blockers for 3 months and in 84%–87% of subjects who used them for 12 months. A third study stated that most patients whose symptoms worsened after discontinuation wished to restart their medication rather than undergo surgery.[27]

### Adverse events

Nine studies provided no data on AEs during discontinuation, or if they did, provided data without a clear indication of the treatment group.[24] Another study only reported AEs during follow-up for those who discontinued alpha-blockers.[19] The six remaining studies reported 49 AEs in 363 patients who discontinued alpha-blockers and 58 AEs in 370 patients who continued to use alpha-blockers.[12–15 21 22] The AE rates in patients who discontinued or continued alpha-blockers were 0.13 and 0.15 per 1000 patient-days, respectively. The pooled data showed no risk difference for AEs when discontinuing or continuing either monotherapy (risk difference=−0.01; 95% CI −0.08 to 0.07) or combination therapy (risk difference=−0.03; 95% CI −0.07 to 0.01, figure 5).

Respiratory tract infection and urinary retention were the two most common AEs after discontinuing alpha-blockers (11 studies in total), being reported in 1%–4% of patients[12 21] and in 1%–3% of patients,[12 22] respectively. The incidence of these AEs did not differ between groups.

## DISCUSSION

The results of this systematic review indicate that discontinuing alpha-blocker monotherapy leads to a worsening of clinical symptoms and a decrease of urinary flow rates in the short term (3–6 months) compared with continuing

## Adverse events

| Study or Subgroup | Discontinuation | | Continuation | | Weight | Risk Difference M-H, Random, 95% CI |
|---|---|---|---|---|---|---|
| | Events | Total | Events | Total | | |
| **Monotherapy** | | | | | | |
| Debruyne et al | 39 | 83 | 40 | 84 | 5.0% | −0.01 [−0.16, 0.15] |
| Fabricius | 0 | 14 | 0 | 13 | 6.5% | 0.00 [−0.13, 0.13] |
| Kaplan et al | 2 | 26 | 2 | 27 | 5.7% | 0.00 [−0.14, 0.15] |
| Yanardag et al | 2 | 26 | 3 | 31 | 5.4% | −0.02 [−0.17, 0.13] |
| **Subtotal (95% CI)** | | 149 | | 155 | 22.6% | −0.01 [−0.08, 0.07] |
| Total events | 43 | | 45 | | | |

Heterogeneity: Tau² = 0.00; Chi² = 0.06, df = 3 (P = 1.00); I² = 0%
Test for overall effect: Z = 0.15 (P = 0.88)

| **Combination therapy** | | | | | | |
|---|---|---|---|---|---|---|
| Barkin et al | 4 | 148 | 10 | 149 | 50.2% | −0.04 [−0.09, 0.01] |
| Matsukawa | 2 | 66 | 3 | 66 | 27.2% | −0.02 [−0.08, 0.05] |
| **Subtotal (95% CI)** | | 214 | | 215 | 77.4% | −0.03 [−0.07, 0.01] |
| Total events | 6 | | 13 | | | |

Heterogeneity: Tau² = 0.00; Chi² = 0.37, df = 1 (P = 0.54); I² = 0%
Test for overall effect: Z = 1.59 (P = 0.11)

| **Total (95% CI)** | | 363 | | 370 | 100.0% | −0.03 [−0.06, 0.01] |
|---|---|---|---|---|---|---|
| Total events | 49 | | 58 | | | |

Heterogeneity: Tau² = 0.00; Chi² = 0.86, df = 5 (P = 0.97); I² = 0%
Test for overall effect: Z = 1.47 (P = 0.14)
Test for subgroup differences: Chi² = 0.39, df = 1 (P = 0.53), I² = 0%

**Figure 5** Forest plot of AEs after discontinuing or continuing alpha-blockers. AEs, adverse events.

van der Worp H, et al. BMJ Open 2019;9:e030405. doi:10.1136/bmjopen-2019-030405

therapy. However, after 1 year, no differences were found in these or other outcomes. Discontinuing the alpha-blocker after combination therapy had no significant effects on outcomes in either the short or the long term.

The worsening of symptoms over the short term after stopping monotherapy was probably relevant to clinical practice. The reported differences in the IPSS between groups exceeded the MCID of 2.7 points.[11] The difference in Q-max was also clinically relevant, exceeding the MCID of 2 mL/s after 3 months (between-group difference, 2.59 mL/s),[3] but not after 6 months (1.79 mL/s). One might argue about the relevance of this outcome for patient, as men will not be able to notice a difference in flow rate at these values. The difference of 0.42 point in the QoL scores at 6 months after discontinuing combination therapy remained below the MCID of 0.5 point.[3] Although no MCID was available for PVR, we do not think that the reported mean difference of 10 mL after 3 months was clinically relevant.

The worsening of symptoms noted by 3–6 months after discontinuing monotherapy was larger than the reported improvement of symptoms after initiating therapy, which was reported to be 2.55 points (95% CI, 1.92 to 3.17) based on 12 RCTs with a total of 9335 participants.[3] The magnitude of change in the present review may have been influenced by the lack of blinding of both patients and assessors in many of the studies, which will have favoured the continuation groups. Men in these studies who had no clear symptom improvements are likely to have dropped out before the discontinuation phase, so the participants subsequently included in the discontinuation trials will generally have had larger treatment effects and larger changes after discontinuation. The outcomes after 12 months relied on data from a single study on discontinuing doxazosin (not a controlled-release version),[16] which has a lower efficacy than other alpha-blockers. This might explain the lack of any meaningful long-term impact.

Among patients receiving combination therapy, outcomes were not significantly different between those discontinuing and continuing alpha-blockers. Although 5-alpha-reductase inhibitors have no significant impact on LUTS severity after treatment initiation,[28 29] their continuation seems to be protective against symptom worsening after discontinuing alpha-blockers.

The results for restarting a discontinued alpha-blocker were heterogeneous, ranging widely from 0% to 49%. These conflicting findings can be explained by the differences in instructions given to patients in these studies. Indeed, participants in some studies received explicit instructions about when to restart therapy, whereas in other studies, no instructions were given. Also, subjects in cohort studies who volunteered to discontinue therapy may have had greater freedom to restart therapy than those participating in an RCT.

It was also shown that discontinuing alpha-blockers did not result in more AEs, including acute urinary retention.[30] Equally, continuation was not associated with more AEs, with neither dizziness nor orthostatic hypotension

being more common.[4] This may be explained by subject dropout due to AEs before entering the discontinuation phase. The number of patients reporting AEs in the included studies was, however, too small to draw meaningful conclusions regarding AEs.

Interpretation of our findings is hampered by some limitations. For example, the limited numbers of studies and large amount of statistical heterogeneity limited data pooling. Heterogeneity, especially on IPSS outcomes after 6 months could be explained by differences in alpha-blockers studied and baseline symptom severity differences ranging from 12 to 19 in the included studies. The limited number of RCTs also precluded sensitivity analyses, and subgroup analyses, that were planned in the original review protocol. Another limitation related to the limited number of studies is the reduction in statistical power. It has been shown that at least five studies have to be pooled to achieve a greater power than the original studies independently.[31] So, our results could also be subject to type I error. In addition, some studies gave unclear data about treatment compliance or only presented per protocol analyses, which may have led to bias (eg, dropout due to severe complaints) and loss of generalisability. Another issue is that all studies were performed in secondary or tertiary care settings. This is important if we consider that in some countries, most men with LUTS are treated in primary care. The high risk of bias, which was noted for all but one outcome, also hampers the interpretation of our findings. Finally, two of the trials of combination therapy compared discontinuing alpha-blockers and 5-alpha-reductase inhibitors, but the others compared discontinuing and continuing only the alpha-blocker.[23 25]

No firm conclusions can be drawn from this review because of the low quality of the available evidence. Overall, the data suggest that there is a short-term clinical worsening of LUTS after discontinuing alpha-blocker monotherapy, as assessed by symptom scores and urinary flow rates, but that this does not increase the risk of a complicated symptom course.

Patients frequently discontinue alpha-blocker treatment in clinical practice. We have recently shown that men who continue to use alpha-blockers are typically unconcerned about stopping that therapy if advised to do so by a doctor.[32] The present review also provides evidence that the magnitude of symptom deterioration is limited, indicating that physicians can change their prescribing policy without risking harm. Indeed, the alternative approach may promote unnecessary polypharmacy, which is especially relevant in vulnerable groups. Active follow-up should then be used to monitor the need to restart alpha-blockers if symptoms worsen.

Our findings support the existing EAU guidance to consider discontinuing alpha-blockers in patients receiving combination therapy for 6 months.[1] Unfortunately, because the studies in this review were only performed in secondary care, we cannot give firm support for the recommendation of the Dutch GP guideline

to review therapy after 3–6 months in primary care.[2] Symptom levels before treatment are generally lower in primary care, where conditions are typically less severe than in secondary care. Although the data from this review may be applicable to primary care, further efficacy studies and discontinuation trials are needed to assess the outcomes specific to this setting.

**Acknowledgements** We thank Dr Robert Sykes (www.doctored.org.uk) for providing editorial services.

**Contributors** The corresponding author attests that all listed authors meet authorship criteria and that no others meeting the criteria have been omitted. MHB had the idea for the article. Acquisition of the data was done by IH, LK, HvdW and MHB. Analysis and interpretation of the data was done by HvdW, MHB, PJ, YL-vL and MGS. HvdW and MHB wrote the manuscript. All authors critically reviewed the manuscript.

**Funding** The authors have not declared a specific grant for this research from any funding agency in the public, commercial or not-for-profit sectors.

**Competing interests** None declared.

**Patient consent for publication** Not required.

**Provenance and peer review** Not commissioned; externally peer reviewed.

**Data availability statement** Data are available on reasonable request.

**ORCID iDs**
Henk van der Worp http://orcid.org/0000-0001-5545-4155
Marco H Blanker http://orcid.org/0000-0002-1086-8730

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
