## [Reviewer comments · BMJ Open]

ARTICLE DETAILS

TITLE (PROVISIONAL)	Discontinuation of alpha-blocker therapy in men with lower urinary tract symptoms: A systematic review and meta-analysis
AUTHORS	van der Worp, Henk; Jellema, Petra; Hordijk, Ilse; Lisman-van Leeuwen, Yvonne; Korteschiel, Lisa; Steffens, Martijn; Blanker, Marco

VERSION 1 – REVIEW

REVIEWER	Jae Hung Jung Department of Urology, Yonsei University Wonju College of Medicine
REVIEW RETURNED	23-Apr-2019

GENERAL COMMENTS	Thank you very much for submitting your manuscript. Unfortunately, there are a few that need to be revised. 1. Abstract: I think that there is no guidance for abstract of systematic review in BMJ open. I would recommend using the template of Cochrane Review or Euro Urol. In addition, please clarify that author included the studies regardless of the design and only accessed RoB for the studies with comparison. Also, please describe the outcomes in an order described in 'outcome measures' section. 2. Method 2.1. I would ask author to avoid the term 'non-randomized controlled trials'. It may cause confusion for the readers. I think just use the term non randomized trial for all type of studies with comparison apart from randomized controlled trial. However, quasi-randomized trials can be treated as RCTs. 2.2. Search strategy have to be same in protocol and manuscript. However, author omitted a few entry terms for alpha blockers. In addition, please clarify the search strategy about where the terms came from (e.g. [MeSH] or [Ti/Ab]). 2.3. Why author did not assess the risk of bias for non randomized controlled trials? Newcastle–Ottawa scale or ROBINS-I can be used for this study design. I think author should use these for non-randomized controlled trials (author's definition) instead of Cochrane tool. 2.4. Author said "For AEs, we also calculated the rate of AEs per 1,000 patient-days, based on the sample sizes and follow-up times", however, I could not find the results related to this. If there was no data, please remove this sentence. 2.5. If there is heterogeneity between the studies, we should use random effect model. However, author did use the random effect model if I² was < 40%(not significant heterogeneity). In addition, I would ask author to use 'less than' or 'greater than' instead of mathematical symbol (e.g. <). 3. Results
--

	3.1. Risk of bias: please see above. In addition, please describe the details in the main text for each domain. 3.2. Why author did not pool the results from monotherapy and combination therapy. If concomitant intervention was the same in the experimental and comparator groups (e.g. finasteride), we can pool the data, I think. 3.3. According to AMSTAR-2 guidance, we should describe the results from RCTs and NRCTs separately. 3.4. I would recommend changing the outcome, 'adverse event' to 'overall adverse event' as author did included all type of adverse event regardless of relationship with the drugs. In addition, how about listing the adverse events that were reported in relevant study in the table? 3.5. While author planned to perform subgroup and sensitivity analysis (see the protocol), there were no explanation for the analyses. Only the subgroup mono vs combination was reported. 3.6. Do we need the results with regard to outcome 'average flow rate'? It may not be patient important outcome. Maybe, redundant outcome of 'Qmax'. 3.7. Would you please describe the details related to publication bias in this section as well as in the method section? 4. Discussion 4.1. Author use the definition of MCID in discussion for the first time. Would you please pre-specify the MCID for each outcome in Method section? In addition, please give some explanation with regard to MCID. 4.2. Need some discussion which caused heterogeneity between the studies.
--	--

REVIEWER	Hongjun Li Peking Union Medical College Hospital Beijing, China
REVIEW RETURNED	30-Apr-2019

GENERAL COMMENTS	The current meta-analysis focused on an important issue: discontinuation of alpha-blockers in patients with LUTS. Studies concerning monotherapy (alpha-blocker) or combined therapy (alpha-blocker+5-alpha-reductase inhibitors) were enrolled. Results of the meta-analysis revealed that Discontinuing alpha-blocker monotherapy leads to a worsening compared with continuing therapy. Discontinuing the alpha-blocker after combination therapy had no significant effects on outcomes in either the short or long term. The design is good and the study is of great importance. However, as mentioned in the limitation part of this study. The risk of bias was high. I have several suggestions for the study. 1, the dosage of the monotherapy or combined therapy should be mentioned in the table or in the text. 2, the authors say that sample size is small. I suggest they can perform Trial sequential analysis to get a more convincing result. 3, actually, various approaches have been used for BPH/LUTS. I am not sure whether combined therapy only consist of AB+5-Alpha-reductase?
---

REVIEWER	Stephen Huang The University of Sydney Australia
-----------------	--

REVIEW RETURNED	31-Jul-2019
-----------------	-------------

GENERAL COMMENTS	van der Worp and colleagues submitted their meta-analysis reviewing the effects of discontinuation of alpha-blocker therapy in men with lower urinary tract symptoms. They included 16 articles which were divided into monotherapy and combination therapy. The primary objectives were multiple symptoms scores, including IPSS, Q-rate, PVR and QoL, and secondary objectives were restarting of alpha-blocker therapy and the adverse events incidence rate. The main findings were a deterioration of symptoms at 3 and/or 6 months for those studies using monotherapy. This is a well-written meta-analysis, but was limited by the number of relevant studies for each outcomes. The manuscript can be improved by addressing a few statistical points:  • Please state which method was used for determination of between-study variability (in RevMan). This is important for reproducibility research, and also has implications of the number of studies required to detect a meaningful difference (power) [Guolo and Varin, Stat Meth Med Res 2017; 26:1500-1518]. • Another related question was the limited number of studies available in some outcomes. Most analyses in this study were based on a small number of studies (2 to 3 studies). This was not the fault of the authors, and the authors already pointed out some limitations of the small number of studies (p. 13 – 14). However, the authors should also point out the higher risk of type I error (reduced power) [see Jackson & Turner, Res Syn Meth 2017; 8:290-302]. • The decision to “pool” (“combine” is a better word) RCTs and NRCTs is usually based on risk of bias analysis, similarities of different studies and so on [e.g. Wells et al, Res Syn Meth 2013; 4:63-77]. The authors used heterogeneity ($I^2 < 0.4$) as the basis for making such decision (p. 6), this should be justified given that I^2 may not be necessarily related to study characteristics and quality. Although the authors pointed out that “NRCT data have introduced selection bias” (p. 13-14), the implications on the results should also be discussed. Will combining studies based on I^2 itself be considered as a kind of “selection bias”? • Since the authors looked at the outcomes at 3, 6 and 12 months, the authors should also explain why longitudinal meta-analysis was not used [see for example Musekiwa et al, PLOS one 2016; 11:e0164898]. The authors should be careful not to treat the results as if it was a longitudinal meta-analysis. For example, a false sense of longitudinal effect was perceived in the Discussion (e.g. paragraph 1, and the use of “3-6 months” rather than “3 and 6 months”).
--

VERSION 1 – AUTHOR RESPONSE

Reviewer: Jae Hung Jung

Thank you very much for submitting your manuscript. Unfortunately, there are a few that need to be revised.

1. *Abstract: I think that there is no guidance for abstract of systematic review in BMJ open. I would recommend using the template of Cochrane Review or Euro Urol. In addition, please clarify that author included the studies regardless of the design and only accessed RoB for the studies with comparison. Also, please describe the outcomes in an order described in 'outcome measures' section.*

>> BMJ open asks authors to follow reporting guidelines, in this case the PRISMA statement. Therefore we used the 'PRISMA for abstracts' statement paper (Beller EM, Glasziou PP, Altman DG, et al. *PLoS Medicine*. 2013;10(4):e1001419). We added information about eligibility and RoB to the abstract and changed the order of outcomes to match the order in the manuscript as suggested.

Abstract

Eligibility criteria: We selected studies **regardless of study design** in which men were treated with an alpha-blocker for at least 3 months and in which the effects of alpha-blocker discontinuation were subsequently studied.

Risk of bias: Two reviewers independently assessed the risk of bias **for the controlled studies only**, using the Cochrane Collaboration's tool for assessing risk of bias.

2. Method

2.1. *I would ask author to avoid the term 'non-randomized controlled trials'. It may cause confusion for the readers. I think just use the term non randomized trial for all type of studies with comparison apart from randomized controlled trial. However, quasi-randomized trials can be treated as RCTs.*

>> We thank the reviewer for this comment and understand the confusion that this term might cause. We have made the suggested changes in the methods section and Table 1 (column 'design'):
'For the primary objective, only randomized controlled trials (RCTs, **including quasi-randomized trials**) and non-randomized trials (NRTs) that compared alpha-blocker discontinuation to continuation were selected.'

We also updated the overview of study designs in the results section:

Nine studies (772 participants) reported discontinuing alpha-blocker monotherapy: two double-blind RCTs,[11, 12] two open-label RCTs,[13, 14] one NRCT,[15] and four uncontrolled studies.[16-19] Six studies (980 participants) reported discontinuing alpha-blockers used in combination therapy: one double-blind RCT,[20] four open-label RCTs,[21-24] and one uncontrolled study.[25]

2.2. *Search strategy have to be same in protocol and manuscript. However, author omitted a few entry terms for alpha blockers. In addition, please clarify the search strategy about where the terms came from (e.g. [MeSH] or [Ti/Ab]).*

>>We have synchronized the search strategy in the manuscript to match the protocol that was followed (supplementary file 1). If applicable, [MeSH] was added. For all other terms, no restrictions were applied.

Supplementary file 1

(Adrenergic alpha blockers [MeSH] OR Adrenergic alpha blockers OR alpha blockers OR doxazosin OR terazosin OR silodosin OR tamsulosin OR alfuzosin) AND (LUTS [MeSH] OR Prostatic Hyperplasia [MeSH] OR LUTS OR BPH OR lower urinary tract symptoms OR benign prostate hypertrophy OR benign prostate enlargement) AND (discontinu* OR interrup* OR cessa* OR stop* OR withdra* OR intermit*).

2.3. *Why author did not access the risk of bias for non randomized controlled trials? Newcastle–Ottawa scale or ROBINS-I can be used for this study design. I think author should use these for non-randomized controlled trials (author's definition) instead of Cochrane tool.*

>> We acknowledge that assessing the risk of bias for non-randomized controlled trials may be relevant. We have assessed the RoB of all trials with the Cochrane tool (Higgins). We classified three studies as non-randomized, but two of these were quasi-randomized (Kaplan et al. and Liaw & Kuo). We now classified these more correctly as randomized studies (following comment 2.1). This leaves one non-randomized study (Gerber et al.) for which RoB has been assessed with the Cochrane tool.

For two reasons we prefer to assess this study with the Cochrane tool. First we think that rating one study with a different tool complicates comparison between studies. Secondly, the main difference between ROBINS and the Cochrane tool is that the former accounts for bias in measurement of exposure and outcome which can be present in observational studies. This however, is not an issue in the study by Gerber because this trial was controlled. The main source of bias in this study is selection bias because subjects could choose the intervention. Assessment of this bias is included in the Cochrane tool though.

No changes were made to the manuscript. We hope that you will agree with this.

2.4. Author said "For AEs, we also calculated the rate of AEs per 1,000 patient-days, based on the sample sizes and follow-up times", however, I could not find the results related to this. If there was no data, please remove this sentence.

>> In our manuscript, we have presented these results as the final part of the Results section. (page 12). Here we have mentioned: 'The AE rates in patients who discontinued or continued alpha-blockers were 0.13 and 0.15 per 1,000 patient-days, respectively.'

No changes were made to the manuscript.

2.5. If there is heterogeneity between the studies, we should use random effect model. However, author did use the random effect model if I2 was < 40%(not significant heterogeneity). In addition, I would ask author to use 'less than' or 'greater than' instead of mathematical symbol (e.g. <).

>> Although random effects models are indicated in the case of high heterogeneity, they can also be used when little heterogeneity is present. The Cochrane explicitly warns against choosing between fixed and random effects models based on statistical test for heterogeneity. Given that there is some variation between the studies in for example the exact type of medication or baseline symptoms we think that using a random effects model is more appropriate.

No changes were made to the manuscript. We hope that you will agree with this.

3. Results

3.1. Risk of bias: please see above. In addition, please describe the details in the main text for each domain.

We have added the five different domains in the Methods section. As the Cochrane tool is widely known, we feel that providing more details is not needed. If you prefer more details, however, we will add more information.

Risk of bias was described for the **five domains (selection, performance, attrition, reporting, and other)** and summarized across studies and outcomes.[7]

3.2. Why author did not pool the results from monotherapy and combination therapy. If concomitant intervention was the same in the experimental and comparator groups (e.g. finasteride), we can pool the data, I think.

>> We decided not to pool the data of monotherapy and combination we feel that both treatment regimes differ and are not comparable beforehand. This clinical heterogeneity is the main reason not to pool these two treatments. Therefore, we have chosen to perform separate analyses, as presented.

The results do support this decision as discontinuation of combination therapy does seem to have a smaller effect compared to discontinuation of monotherapy. We had already discussed these differences in the discussion section.

3.3. According to AMSTAR-2 guidance, we should describe the results from RCTs and NRCTs separately.

>> We agree that results from RCTs and NRCTs should be described separately. Please note that we included only one NRCT (Gerber et al) and did not combine the outcomes of RCTs and NRCTs anywhere in our systematic review. Therefore, we did not change the manuscript at this point.

3.4. *I would recommend changing the outcome, 'adverse event' to 'overall adverse event' as author did included all type of adverse event regardless of relationship with the drugs. In addition, how about listing the adverse events that were reported in relevant study in the table?*

>> It is unclear to us what the reviewer means with this comment. We have summarized the AEs presented in the included studies. We were able to pool data from six studies in which AEs were reported for the two treatment groups (continuation versus discontinuation). Especially because the incidence of AEs did not differ between groups, we doubt if presenting all AEs in a separate table truly adds to this paper. We decided not to change the manuscript at this point. We hope you agree with that.

3.5. While author planned to perform subgroup and sensitivity analysis (see the protocol), there were no explanation for the analyses. Only the subgroup mono vs combination was reported.

Indeed, we had planned more subgroup and sensitivity analyses in our protocol, following general guidelines for meta-analyses. As the number of included studies was small for the main analyses, performing these subgroup analyses and sensitivity analysis would have led to single study comparisons. Therefore, we have refrained from performing such analyses.

>> We have added the following to the already mentioned information in the discussion paragraph: . The limited number of RCTs also precluded sensitivity analyses, **and subgroup analyses, that were planned in the original review protocol.**

3.6. Do we need the results with regard to outcome 'average flow rate'? It may not be patient important outcome. Maybe, redundant outcome of 'Qmax'.

>> We agree with the reviewer that flow rate may not be patient relevant. In the urological literature, however, flow rate is a commonly reported outcome, when evaluating treatment in male LUTS. In fact, the difference shown in our review is defined as clinically relevant, as it exceeded the MCID for this outcome. We have added the following to the discussion (second paragraph): **One might argue about the relevance of this outcome for patient, as men will not be able to notice a difference in flow rate at these values.**

3.7. Would you please describe the details related to publication bias in this section as well as in the method section?

>> Thank you for this suggestion. We have added the following to the methods section and result section.

In our protocol we had defined the following: In case at least 10 studies are included in the meta-analysis we will construct funnel plots to ascertain graphically the existence of publication bias.

We have added the following to the Materials and methods section: **To ascertain graphically the existence of publication bias, the construction of funnel plots was planned in case at least 10 studies were included.**

We have added the following to the results section: **We did not construct funnel plots to ascertain the existence of publication bias graphically, as the number of included studies in each meta-analysis was less than 10.**

4. Discussion

4.1. Author use the definition of MCID in discussion for the first time. Would you please pre-specify the MCID for each outcome in Method section? In addition, please give some explanation with regard to MCID.

>> We have added the following to the methods section:

The following cut-off values were used to define the minimal clinical important difference (MCID): 2.7 points for IPSS,[27] 2mL/s for Q-max,[3] and 0.5 points for the IPSS-QoL scores.[3]

The MCID is the smallest change in a treatment outcome that an individual patient would identify as important and which would indicate a change in the patient's management.

4.2. Need some discussion which caused heterogeneity between the studies.

>> heterogeneity was specifically seen for the analyses on IPSS scores at six months, and could be explained by differences in the type of alpha-blocker under study and baseline symptom severity ranging from 12 to 19 points.

We have added the following to the discussion paragraph: **Heterogeneity, especially on IPSS outcomes after 6 months could be explained by differences in alpha-blockers studied and baseline symptom severity differences ranging from 12 to 19 in the included studies.**

Reviewer: 2 Hongjun Li

The current meta-analysis focused on an important issue: discontinuation of alpha-blockers in patients with LUTS. Studies concerning monotherapy (alpha-blocker) or combined therapy (alpha-blocker+5-alpha-reductase inhibitors) were enrolled. Results of the meta-analysis revealed that Discontinuing alpha-blocker monotherapy leads to a worsening compared with continuing therapy. Discontinuing the alpha-blocker after combination therapy had no significant effects on outcomes in either the short or long term.

The design is good and the study is of great importance. However, as mentioned in the limitation part of this study. The risk of bias was high.

I have several suggestions for the study.

1, the dosage of the monotherapy or combined therapy should be mentioned in the table or in the text.

>> Thank you for this valuable suggestion. We have included the information about dosage in Table 1.

2, the authors say that sample size is small. I suggest they can perform Trial sequential analysis to get a more convincing result.

>> The Cochrane Collaboration argues against the use of Trial sequential analysis (https://methods.cochrane.org/sites/default/files/public/uploads/tsa_expert_panel_guidance_and_recommendation_final.pdf). Cochrane advises that authors should interpret evidence on the basis of the estimated magnitude of the effect of intervention and its uncertainty (usually quantified using a confidence interval), rather than focusing primarily on the rejection of the null hypothesis of no treatment effect. In our manuscript, we have compared the outcomes with the MCID. As such we have followed the Cochrane guideline. We therefore decided not to add the requested analysis. We hope you will agree on this.

3, actually, various approaches have been used for BPH/LUTS. I am not sure whether combined therapy only consist of AB+5-Alpha-reductase?

>> Other combinations of drug treatments have been studied as well. As for primary care, alpha-blocker treatment is the primary choice, and combination with anticholinergics are not advocated, we decided to focus only on alpha-blockers and 5ARI. As this focus was taken from the beginning, we cannot add other comparisons in this stage. No changes were made at this point.

Reviewer: 3, Stephen Huang

van der Worp and colleagues submitted their meta-analysis reviewing the effects of discontinuation of alpha-blocker therapy in men with lower urinary tract symptoms. They included 16 articles which were divided into monotherapy and combination therapy. The primary objectives were multiple symptoms scores, including IPSS, Q-rate, PVR and QoL, and secondary objectives were restarting of

alpha-blocker therapy and the adverse events incidence rate. The main findings were a deterioration of symptoms at 3 and/or 6 months for those studies using monotherapy. This is a well-written meta-analysis, but was limited by the number of relevant studies for each outcomes.

The manuscript can be improved by addressing a few statistical points:

- *Please state which method was used for determination of between-study variability (in RevMan). This is important for reproducibility research, and also has implications of the number of studies required to detect a meaningful difference (power) [Guolo and Varin, Stat Meth Med Res 2017; 26:1500-1518 PubMed].*

>> Thank you for this suggestion. We have included the following information to the methods section: 'We calculated the risk difference and 95% confidence intervals (CIs) for dichotomous variables (inverse-variance method) and the mean differences (MD) with 95% CIs for continuous variables (Mantel-Haenszel method).'

- *Another related question was the limited number of studies available in some outcomes. Most analyses in this study were based on a small number of studies (2 to 3 studies). This was not the fault of the authors, and the authors already pointed out some limitations of the small number of studies (p. 13 – 14). However, the authors should also point out the higher risk of type I error (reduced power) [see Jackson & Turner, Res Syn Meth 2017; 8:290-302 PubMed].*

>> We have now added this point to the discussion of the limitations:

'Another limitation related to the limited number of studies is the reduction in statistical power. It has been shown that at least five studies have to be pooled to achieve a greater power than the original studies independently.[REF] So, our results could also be subject to Type I error.' We have added the reference to Jackson & Turner as well.

- *The decision to “pool” (“combine” is a better word) RCTs and NRCTs is usually based on risk of bias analysis, similarities of different studies and so on [e.g. Wells et al, Res Syn Meth 2013; 4:63-77 PubMed]. The authors used heterogeneity ($I^2 < 0.4$) as the basis for making such decision (p. 6), this should be justified given that I^2 may not be necessarily related to study characteristics and quality. Although the authors pointed out that “NRCT data have introduced selection bias” (p. 13-14), the implications on the results should also be discussed. Will combining studies based on I^2 itself be considered as a kind of “selection bias”?*

>> We agree with the reviewer that the decision to pool data is a two-phase decision. First, we have decided if we felt that data could be pooled, based on clinical comparison of studies (type of drug used, patient characteristics). Next, the statistical heterogeneity. In the manuscript, we did not describe this process adequately. In our protocol we had mentioned: *We will assess clinical heterogeneity by checking the characteristics of participants and interventions, and statistical heterogeneity by visual inspection the outcomes in the forest plots and by the formal statistical test for heterogeneity (i.e. the I^2 statistic). In case two or more studies are available and the studies are sufficiently homogeneous in terms of participants, interventions and outcomes to provide a meaningful summary we will perform meta-analyses using a random effects model. Otherwise we will summarize the results descriptively. In case at least 10 studies are included in the meta-analysis we will construct funnel plots to ascertain graphically the existence of publication bias.* We followed the protocol, but didn't mention that in the manuscript.

We have added the first step of our decision on pooling to the methods section (**Clinical heterogeneity was assessed by checking the characteristics of participants and interventions.**), and the following information to the results section: **No studies were excluded from pooling based on statistical heterogeneity.**

We agree that using I^2 statistics itself could also introduce bias, but in our study this was not the case. Notably, we did not pool NRCT information with RCT outcomes.

• Since the authors looked at the outcomes at 3, 6 and 12 months, the authors should also explain why longitudinal meta-analysis was not used [see for example Musekiwa et al, PLOS one 2016; 11:e0164898]. The authors should be careful not to treat the results as if it was a longitudinal meta-analysis. For example, a false sense of longitudinal effect was perceived in the Discussion (e.g. paragraph 1, and the use of “3-6 months” rather than “3 and 6 months”).

>> We had no aim to describe longitudinal outcomes. Different studies used different periods of follow-up (listed in Table 1), which we have combined in separate analyses. We agree that the results of our study should not be interpreted as longitudinal meta-analyses. We have changed the phrases that gave a false sense of longitudinal effect, which is not justified base on the used analysis, as follows:

Discussion section:

*The results of this systematic review indicate that discontinuing alpha-blocker monotherapy leads to a worsening of clinical symptoms and a decrease of urinary flow rates in the short-term (3–6 months) compared with continuing therapy. However, **in the analyses on one year follow-up**, no differences were found for these or other outcomes.*

Next, we have added this warning to the limitations section of the discussion: **Notably, the outcomes of our review cannot be interpreted as longitudinal analyses. Each period of follow-up was studied separately, including different studies.**

VERSION 2 – REVIEW

REVIEWER	Hongjun Li Peking Union Medical College Hospital, Beijing, China
REVIEW RETURNED	09-Oct-2019

GENERAL COMMENTS	I have no further comments for this manuscript
--

REVIEWER	Stephen Huang Nepean Hospital The University of Sydney Australia
REVIEW RETURNED	14-Oct-2019

GENERAL COMMENTS	The authors have addressed my concerns. However, the abbreviations "NRCT" is still used throughout the manuscript although the authors has changed it to non-randomised trial "NRT" (p.5).
--

VERSION 2 – AUTHOR RESPONSE

Reviewer: 3

The authors have addressed my concerns. However, the abbreviations "NRCT" is still used throughout the manuscript although the authors has changed it to non-randomised trial "NRT" (p.5).

>>We have made the requested change throughout the manuscript.